# Attitude and Purchase Intention to Generic Drugs

**DOI:** 10.3390/ijerph18094579

**Published:** 2021-04-26

**Authors:** Ricardo Arcaro, Cássia Rita Pereira da Veiga, Wesley Vieira da Silva, Claudimar Pereira da Veiga

**Affiliations:** 1Graduate Program in Organizations Management, Leadership and Decision (PPGOLD), Department of General and Applied Administration, Federal University of Parana, Curitiba 80210-170, Brazil; ricardo.arcaro@ufpr.br (R.A.); cassia.veig@gmail.com (C.R.P.d.V.); 2Faculty of Economics, Administration and Accounting, Federal University of Alagoas, Maceió 57072-900, Brazil; wesvsilva@gmail.com

**Keywords:** purchase intention, generic drugs, attitude, consumer behavior

## Abstract

Generic drugs were instituted in 1984 in the United States. Since that time, many studies have been conducted in several countries into consumer attitude and behavior when purchasing generic drugs. Understanding the factors that can influence attitude and purchasing intention in this segment has been a challenge. Thus, this paper aims to present a mapping of the literature on the attitude toward and intention to purchase generic drugs and capture insights that can help define and improve promotional strategies for the use of these products. To identify articles related to the theme, we selected the Web of Science, Science Direct, Scopus, Lilacs, Pubmed Central, Springer, and Embase databases time limited to June 2020, using the keywords “generic drug”, “purchase intention”, and “attitude”. The results indicate that this topic is relatively new, with publications in the leading journals in the area demonstrating its importance. Analysis revealed five strategic insights and showed that the research theme could be grouped into three clusters: (i) consumer attitude and behavior, (ii) perspective of patients and health professionals, and (iii) assessment of the risks associated with generic medications to determine which factors can influence purchase intention, providing decision makers with a broader view with regard to directing public policy strategies in healthcare.

## 1. Introduction

The pharmaceutical industry is one of the players that most invests in research and innovation [1]. In the wake of the Trade Related Aspects of Intellectual Property Rights (TRIPS) (1994) agreement [2], many countries have regulated patent protection for pharmaceutical products, which was instrumental in encouraging investments in clinical studies and research [3]. Most pharmaceutical products have production processes that can be easily copied with lower investments than those required for the original patented product [4]. After the expiration of the exclusivity period guaranteed by the patent [5], generic drug companies increase market competition, which can result [6,7], or not [8], in price reductions and help to ensure access to essential pharmaceutical products [9].

According to the Food and Drug Administration (FDA), a generic drug is identical or bioequivalent to the brand name drug (reference listed drug) in (i) dosage, (ii) safety, (iii) concentration, (iv) route of administration, (v) quality, (vi) performance characteristics, and (vii) intention to use [10]. Generic drugs were instituted at different times in different countries, from 1984 in the United States [11] to 1996 in Italy [12].

The Global Generic Drugs Market (2020) highlights that generic drugs saw a worldwide compound annual growth rate (CAGR) of 8.7% between 2016 and 2020, in a market dominated by many companies such as Mylan, Teva, Novartis, and Sun Pharmaceutical, which together account for 35% of the market share. For companies, understanding consumers’ attitude and intention to purchase pharmaceutical products [13,14] is fundamental [15], and this is also the case when it comes to formulating public health policies and strategies [16].

Attitude towards purchasing generic drugs can be influenced by the perceived quality, product attributes, past experience, and doctors’ recommendations [13]. Product attributes can influence attitude and purchase intention. In this context, we can conclude that understanding the consumer’s journey in purchasing generic drugs related to attitude and purchase intention has been a challenge in the literature [17].

To define purchase intention, this study was based on the definition proposed by [18,19,20]. These authors highlight that purchase intention is classified as one of the components of consumers’ cognitive behavior [18] regarding the means, form, and probability of buying a specific product [18,21]. This study was based on research conducted by [22]. This author explains that purchase attitude is based on determining the behavior for deciding on a future purchase. In this context, purchase attitude includes trust in the product and familiarity related to the consumer’s purchasing power [17]. To this author, in the minds of consumers, the buying attitude is based on the result of (i) cognitive, (ii) emotional, and (iii) affective reactions.

Studies that evaluate perceived risk show that this is a multidimensional construct involving psychological risk, time risk, physical risk, and social risk [23]. Physicians and patients perceive risk in the use of generic drugs because they consider them to not be as safe or effective as the brand name alternatives [24]. Clinical effectiveness was reported by 71.9% of physicians in Saudi Arabia as the most influential factor affecting the prescription of brands over local generic medication [25]. On the other hand, shopping attitude and purchase intentions are sensitive to an economic recession. From the consumer’s point of view, the perceived risk in relation to the use of generic drugs was reduced during periods of economic crisis in Brazil [13].

Attitude and purchase intention for generic drugs have been highlighted in several studies [13,26,27,28,29], revealing that acceptance of generic drugs varies according to the degree of need, which is more favorable for less severe conditions [30]. Studies that evaluate the packaging can help improve information and increase consumer confidence [31]. While the population, in general, has difficulty accepting generic drugs, there seems to be a belief that reference products have better quality and safety when compared to generic drugs [32].

Strategies to accelerate the promotion of generic drugs vary according to public health policies in each country [33]. Some strategies are related to the following:(i)More affordable prices for generic drugs [34].(ii)A legal obligation to prescribe low-cost generic drugs [14].(iii)Creation of computerized systems that assist in the prescription of generic drugs [35].(iv)Policies that facilitate the replacement of reference drugs with generic [36].

However, the increase in the market share of generic drugs remains slow [37]. In this sense, several studies highlight the need to encourage the use of generic drugs. Although it is not easy to maximize the use of generic drugs, this action becomes arduous without the public’s positive perception and meeting their information needs about generic drugs [38,39,40]. Studies indicate that it is essential to increase consumer confidence and knowledge about generic drugs on the market [38].

Generic drugs have increasingly attracted the interest of many stakeholders as a means for patients to receive the same treatment at a lower cost [41]. At the same time, studies show that the growth of this market has been slow [37]. In this context, research into consumer behavior when purchasing generic medications has attracted the interest of public health policymakers, business managers, and academic researchers [14,42], mainly in countries where regulation encourages the production and commercialization of generic drugs, such as India [43] and Brazil [44].

Despite the importance of this theme and the growing number of publications on generic drugs, some controversy remains [38]. Understanding the factors influencing attitude and purchase intention in this segment has been a challenge [13]. Although the important role that the attitude and purchase intent plays can bring insights on maximizing the use of generic drugs, no article has systematically evaluated the “purchase intent”, “attitude”, and “generic drugs” for capturing insights that can assist decision-makers. This systematic mapping of the literature objective was to examine and evaluate studies on the attitude and intention to purchase generic drugs to capture insights that can use direct strategies to promote the use of such products and identify which factors influence consumer purchases. In this context, understanding the attitude and purchase intent and bringing insights that can influence the use of generic drugs is necessary to guide future actions of public policies, education, and practical interventions to maximize the use of generic drugs.

## 2. Materials and Methods

This study used the protocol proposed by [45] and the Preferred Reporting Items for Systematic Reviews and Meta-analyses PRISMA [46] for conducting a systematic mapping of the literature [47,48]. The research was conducted using the Web of Science, Science Direct, Scopus, Lilacs, Pubmed Central, Springer, and Embase databases, time limited to June 2020, using the keywords “generic drug”, “purchase intention”, and “attitude”, as detailed in Table 1 and Figure 1.

Figure 1 shows the search string in the previously selected databases that returned 198 articles, of which 56 were duplicate articles. When applying the screening and eligibility criteria, and selecting only articles available in English, 55 works remained. Finally, the abstract, title, and keywords of the remaining articles were evaluated by two independent reviewers to verify their relevance to the research’s central theme. A third reviewer evaluated any papers flagged in this process. After this screening and eligibility phase, the research proceeded with 13 articles that were analyzed with the aid of bibliometric tools, as well as a qualitative individual analysis to capture strategic insights.

The resulting database was analyzed by VOSviewer software (version 1.6.15) (Centre for Science and Technology Studies, Leiden University, The Netherlands) and Biblioshiny (based on version R 3.6.1, package Bibliometrix version 2.2.1, University of Naples Federico II, Italy). The analysis of the journals was tabulated and evaluated using the Scimago Journal Rank (SJR), while the spectroscopy of the year of publication was performed by analyzing how frequently references were cited in the publications of the research corpus from the year of publication and demonstrated through peaks [46].

To carry out a literature mapping, there is a need to understand which countries make up the corpus of recent research [49]. For this purpose, an analysis of geographical coverage of publications was used with the aid of MapChart (https://mapchart.net/world.html). To identify the researched field’s intellectual structure, co-citation analysis of the references was used with a minimum number of citations for the references. This bibliometric indicator was assessed through the research corpus and demonstrated through networks [50].

The thematic evolution was based on the word network and clustering through the methodology inspired by [51], whereas the three-field plot analysis lists the main items of three distinct fields, both of which are demonstrated by a Sankey diagram [52].

## 3. Results

Table 2 shows a summary of studies of the textual corpus of the research, with 13 articles included in the textual corpus. Of this total, ten articles were relevant to consumer behavior for generic drugs, and five articles were relevant to health professionals. In the studies presented, the most frequently used methodological approach was quantitative, with the application of questionnaires. The table comprised the subject, sampling (*n*), data analysis (type), focus of the study, main findings, and references.

Given the topic’s importance based on the growing number of publications on generic drugs, some questions remain controversial. Therefore, it was necessary to evaluate the literature on attitude and intention to purchase in order to capture insights that can help define strategies to promote the use of generic drugs. As a first step, we analyzed the visibility and prestige of journals based on Scimago quartiles. Table 3 presents the main journals that formed the research corpus, of which 75% are classified in the Q1 quartile of Scimago. It is important to note that there is no concentration of publications on the subject in any one periodical, which demonstrates that the evaluation of the consumer’s purchase behavior regarding pharmaceutical products is of a multidisciplinary nature.

Figure 2 shows the journals that are most frequently cited in the articles that comprise the selected database. The 521 references were evaluated by Rstudio software and show that the Health Policy journal stands out with 22 citations. Following this, the Journal of Consumer Research and Pharmacy World & Science are the journals with the highest number of citations, 15 and 11, respectively.

Figure 3 details the spectroscopy analysis of the year of publication of the references and demonstrates the historical roots of the researched corpus through the relationship between the previously published literature and the most recent literature [62]. Figure 3 is composed of peaks and troughs related to the frequency of citations and reflects the importance of prior awareness for the dissemination of knowledge. The first citation peak of the researched corpus shows the work of [63], which includes three conceptual models of the doctor–patient relationship: “Model of Activity-Passivity”, “Model of Guidance-Cooperation” and “Model of Mutual Participation”. After the 1980s, several peaks were noted, with a greater number of citations between 2009 and 2013, demonstrating that research on consumer attitude and purchase intention for pharmaceutical products represents a theme with a recent knowledge base.

Figure 4 shows the co-citation network of the references considering a minimum of three citations. We found that references from the research corpus can be grouped into three clusters, based on the abstracts read and validated between two authors. In cases of divergence, we included a third reviewer: (i) consumer attitude and behavior (red), (ii) patients and health professionals’ perspective (green), and (iii) assessment of risks associated with generic drugs (blue). Table 4 details the references that are part of each cluster in the correlation network in Figure 4.

Figure 4 shows the co-citation networks with the references with the highest interconnection of the research corpus. Cluster 1 is composed of seven references related to consumers’ attitude and behaviors, with emphasis on the research by [64] that assesses the attitude of consumers towards the use of generic drugs as an alternative to the use of reference pharmaceutical products. In the study by [65], conducted in Portugal, the author presents an argument regarding the underutilization of generic drugs in addition to assessing the attitude of patients and pharmacists in relation to the substitution of branded drugs by generic drugs, whereas [32] assessed the behavior of Japanese consumers with regard to generic drug substitution. Ref. [66] analyzed the intention to purchase generic drugs over the counter (OTC) compared with branded drugs. Ref. [67] examined the perception of consumers regarding the quality of free medicines in India. Ref. [68] assessed the relationship between communication concerning generic drugs and consumer beliefs, while [69] assessed consumers’ perception of the quality of generic drugs.

Cluster 2 is formed by four studies related to the theme of patients and health professionals’ view of generic drugs, with emphasis on the work of [70]. These authors argue about patients’ views and the role of health professionals in the underuse of generic drugs. Ref. [71] evaluates perceptions, knowledge, and attitude towards generic drugs, highlighting the importance of counselling by doctors and pharmacists. In the work of [72], the literature on knowledge, attitude, and opinions of generic drugs was reviewed, highlighting the need for greater communication between health professionals and patients. Finally, [73] drew conclusions about patients’ negative experiences when their drugs were replaced by generics.

Cluster 3 is composed of three references on the theme of risks associated with generic drugs, evidenced in the study by [74] that evaluated doctors’ and consumers’ perceptions regarding substitution by generic drugs. The authors concluded that not all interchangeable drugs are effective and safe. Ref. [26] explored the relationship between consumers’ risk perception versus savings when purchasing a generic drug, and [75] assessed the risk perceived by the consumer associated with acceptance of generic drugs.

The implementation of specific regulations to encourage the production and use of generic drugs around the world occurred at different times and in different ways [76]. Likewise, interest in academic research related to the intention to purchase generic drugs also varied from country to country. Based on the textual corpus, India and the United States are considered mature pharmaceutical markets based on the time since sector-specific regulation was implemented [77]. In the U.S.A., nine out of ten prescriptions filled out by doctors are for generic drugs, increasing market competition and the availability of generic drugs [78]. In turn, India stands out as the largest global supplier of generic drugs, with 40% of demand directed to the United States consumer market [79]. The three-field plot analysis in Figure 5 highlights the differences between the two countries in the topics of interest regarding generic drugs.

Figure 5 correlates the countries of origin of the research with the information from the abstracts and sources cited in a Sankey diagram [52]. The width of the arrows demonstrates the strength of the relationship between the three variables that were surveyed. Research on intention to purchase generic drugs in the United States shows a strong correlation with the words “generic drugs”, “consumers”, and “prescription”, reflecting the country’s concern with its consumer pharmaceutical market and buying behavior and the prescription of generic drugs [80]. The key periodicals are the Journal of Consumer Research, Medical Care, the Journal of Marketing, and the Journal of Applied Psychology.

Research on intention to purchase generic drugs in India has the strongest relationship with the words “health” and “medicines”. As previously described, India is an important producer of generic medicines, but the Indian public have difficulty in accessing medicines, principally for chronic diseases [53]. The journal used for publication of research into this topic was Health Policy and Planning.

In summary, this work evaluated the evolution of consumer attitude and intention to purchase generic drugs over two different time periods due to the fact that there has been an increase in the number of publications since 2013. As shown in Figure 6, the publications before 2013 can be divided into two clusters: decision-making and risks. After 2013, the decision-making cluster can be segmented into generic drugs, medicines, and perceived risks, demonstrating that decision-making is more focused on the product through costs and profit margin [13,54], promotional materials [55], public policies [56], and the quality of generic drugs [42]. Likewise, the risks cluster can be divided into medicines, generic substitution, and patient attitude, which may suggest that patients are suspicious regarding the effectiveness of generic drugs, reinforcing the need to improve communication to health professionals and final consumers [57] and to facilitate access to generic drugs [53].

## 4. Discussion

In this study, we analyzed a literature gap by conducting a systematic mapping of the attitude and intention to purchase generic drugs. We realized that the factors associated with attitude and purchase intention can bring insights in directing strategies to promote the use of such products. There is a need to generate positive perception for doctors, health professionals, and the patient. As far as we know, this is the first study involving a systematic mapping of the literature on the subject.

We identified studies that highlight the challenges involved in implementing policies for the industrialization and commercialization of generic drugs, even though such policies can generate savings for the public health services [38,81]. The systematic mapping of the literature captured five insights that can help guide promotional strategies for the use of generic drugs, as well as provide a useful summary of the current state of research on the topic and the gaps.

The first insight is related to the concentration of the studies. The main journals that published on the topic are classified in the Q1 quartile of Scimago, and there is no concentration of publications, which demonstrates that the evaluation of purchasing behavior based on attitude and intention to buy pharmaceutical products is a discipline which covers several areas. The second insight was related to the frequency of citations of articles since 1980. Given the relevance of the topic, it reflects the importance of prior knowledge for the dissemination of research. It demonstrates that most of the research in this respect is recent, primarily between 2009 and 2013, and highlights an opportunity for additional research. These insights can guide strategies for future research into consumer behavior in the generic drug segment, focusing on the promotion and use of these products.

The main finding of this study is related to the third insight, highlighted in the textual corpus of the research that identified the grouping of three clusters: (i) consumer attitude and behavior, (ii) patients’ and health professionals’ perspective, and (iii) assessment of risks associated with generic drugs, corroborating the study by [38], which emphasizes the main domains that influence the use of generic drugs. Among these domains, the factors related to patients and health professionals influence both the use of and the intention to purchase generic drugs, while the risk associated with generic drugs, in addition to consumers’ attitude and behavior, seems to be inherent to the purchase process. This insight can help guide strategies to understand what the patient really expects from a generic drug. On the other hand, understanding the views of doctors and health professionals is fundamental when it comes to defining more assertive strategies for prescribing and promoting these products [82]. The core of this issue is related to how companies could direct strategies to assess the risks associated with these products, making the product even more reliable for the target audience.

The fourth insight is related to the implementation of specific regulations to encourage the production and use of generic drugs. The review showed that the implementation of such regulations occurred around the globe at different times and in different ways, in accordance with studies by [75]. Likewise, interest in academic research related to the intention to purchase generic drugs also varied from one country to another, especially the United States and India. This insight also highlighted that these two countries are considered mature pharmaceutical markets due to the time that has elapsed since sector-specific regulations were implemented [76].

To illustrate the importance of the generic drug segment, in the USA, 90% of prescriptions are for generics. This factor is intended to boost market competition and the availability of generic drugs [77]. In turn, India stands out as the largest global supplier of generic drugs, with 40% of production directed to the United States consumer market [78]. Figure 5 reinforces the difference between the two countries in the topics of interest related to generic drugs. In the same vein, this insight highlighted the challenges for implementing a generic drug policy that is more specific and that reflects the health systems of each country, and it is necessary to implement measures that can lower barriers among the target audience.

The fifth insight shed light on research on generic drug purchase intent in India (see Figure 5). Research on the intention to purchase generic drugs in this country has the greatest correlation with the words “health” and “medicines”. According to previous research [40], India is an important producer of generic medicines, but its population has difficulty in accessing medicines, especially for chronic diseases.

Likewise, the risks cluster can be divided into (i) medicines, (ii) generic substitution, and (iii) patients’ attitude, which may suggest that patients are suspicious about the effectiveness of generic drugs. This insight suggests the need for improvements in the communication strategy between doctors, health professionals, and the final consumer. This finding corroborates the studies by [57], principally the need for strategies to improve access to generic drugs [53]. Understanding consumer attitude and purchase intention of generic drugs can help companies to achieve a greater market share and influence consumer confidence in the use of these drugs, in addition to being a strategic driver and aid for decision-making.

## 5. Conclusions

The purpose of this article was to conduct a systematic mapping of the literature on attitude and intention to purchase generic drugs, and to capture insights that could help guide strategies to promote the use of such products. The objective was to pinpoint which factors influence consumers’ buying decisions based on evidence from the behavioral areas “purchase intention”, “attitude”, and “generic drugs”, and capture insights that can be used to target strategies to promote the use of generic drugs.

Our results achieved the proposed objective by answering the key research question. We discovered that consumer attitude and intention to purchase generic drugs are related to three main factors, formed by the three clusters of (i) consumer attitude and behavior, (ii) patients’ and health professionals’ views, and (iii) risks associated with generic drugs. We also identified five insights from the current research available on this theme, which can also provide companies marketing generic drugs with strategic direction. The main finding of this study sheds light on how the gap in the relationship between the consumer patient, doctors, public policy makers, and the pharmaceutical industry can be narrowed. On the other hand, there is a need to generate more positive perceptions of generic drugs to increase purchase intention and lower the barriers between different health systems in order to implement policies and promote industrialization, commercialization, and access to generic drugs.

The risks associated with generic drugs in this review were also evidenced in the study by [74] regarding generic drug substitution. It is noteworthy that generic drugs are interchangeable, effective, and safe. This study shows that when consumers consider substituting a reference medicine with the generic equivalent, they will typically weigh up the risks versus the benefits (cost savings) and base their decision on these factors.

Consumer behavior based on attitudes and intention to buy in the generic drug market is a broad and complex topic of which this article did not intend to cover every angle and possibility. Therefore, there is scope for future studies related to the intention to purchase generic drugs, as this is a growing and important market for the pharmaceutical industries, public health policy makers, and consumers. Future research can be conducted with the aim of testing the theory through hypotheses in a meta-analytic study, since a systematic review was conducted without the presence of meta-analyses.

In order to expand the use of generic medicines, the pharmaceutical industry, public health policy makers, doctors, and other health professionals need a better understanding of consumer attitudes and behavior regarding generic drugs. Understanding the key drivers behind consumers’ purchasing intentions will help to guide policies and promotional strategies to lower barriers and encourage a greater uptake of generic drugs.

The main limitation of this study lies in the fact that the available literature is relatively scarce and distributed over few countries. On the one hand, the study has the advantage that it was based on the literature from several continents. On the other hand, there are differences in consumer attitudes and behavior and public health polices from one country and region to the next.

## Figures and Tables

**Figure 1 ijerph-18-04579-f001:**
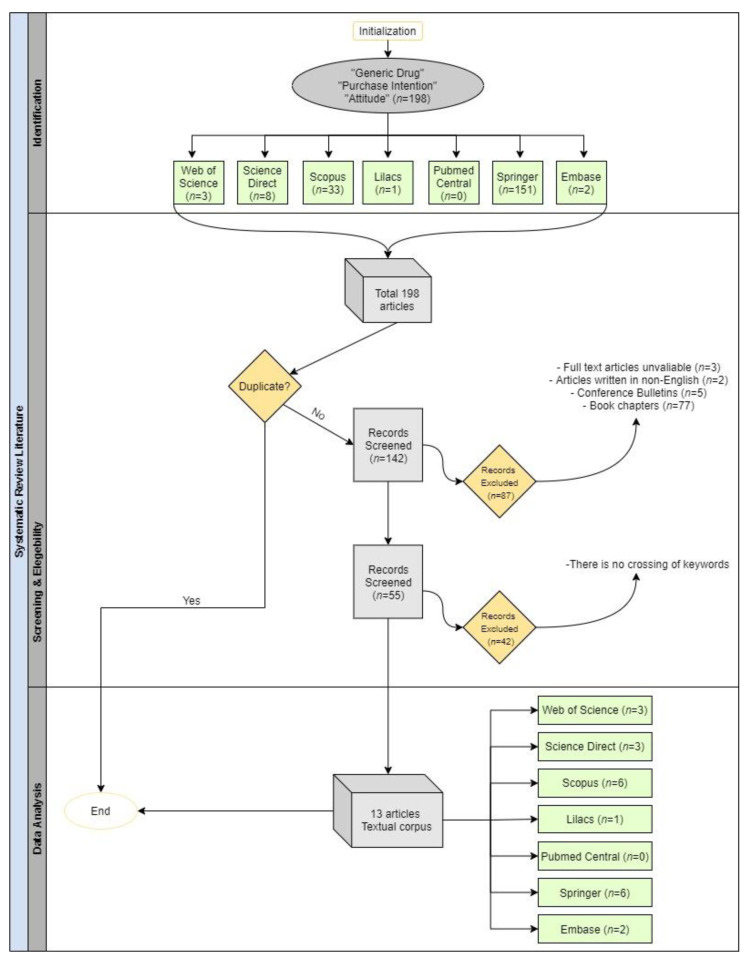
Flowchart of the study selection process following PRISMA guidelines [46].

**Figure 2 ijerph-18-04579-f002:**
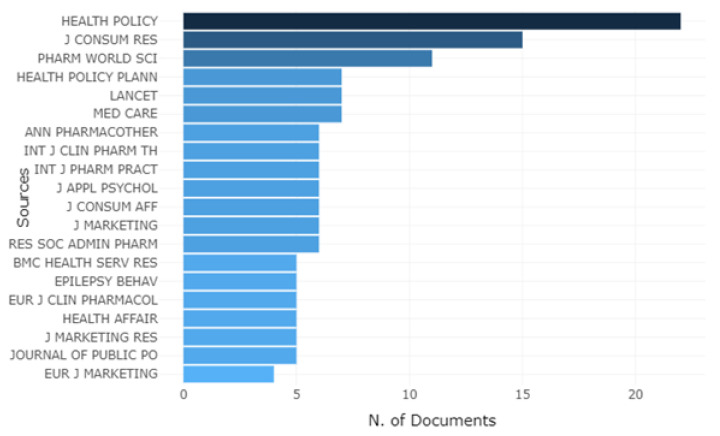
Ranking of the most cited journals.

**Figure 3 ijerph-18-04579-f003:**
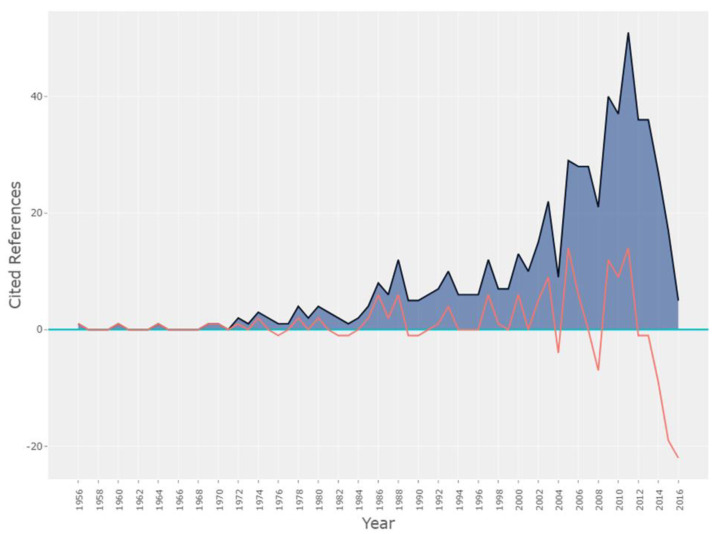
Spectroscopy analysis of the year of publication of the references.

**Figure 4 ijerph-18-04579-f004:**
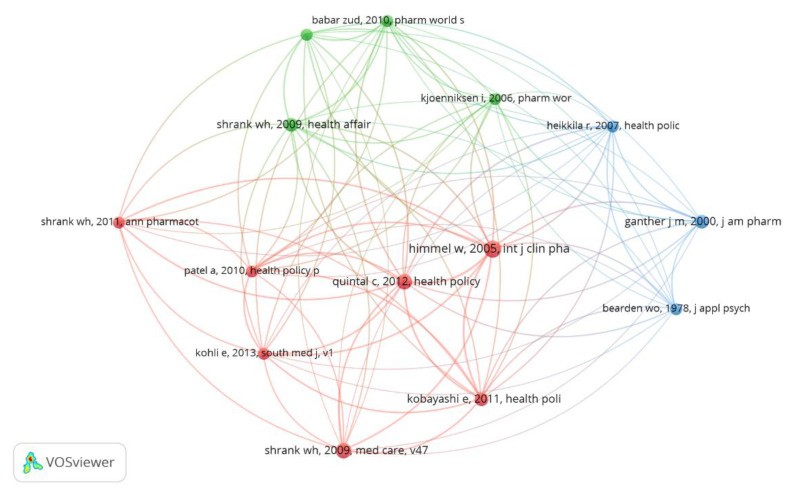
Networks for co-citing references.

**Figure 5 ijerph-18-04579-f005:**
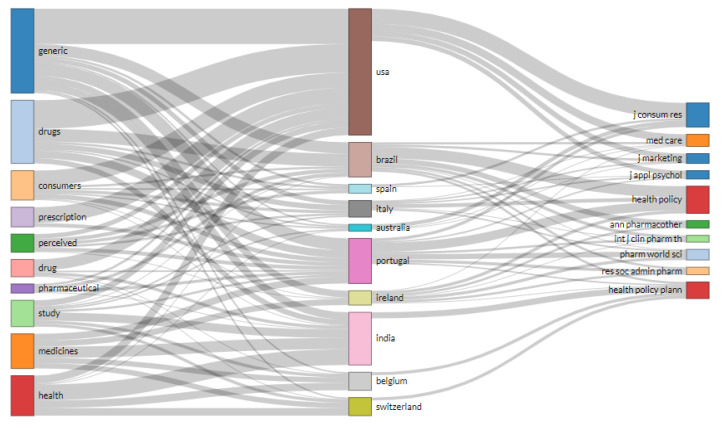
Three-field plots.

**Figure 6 ijerph-18-04579-f006:**
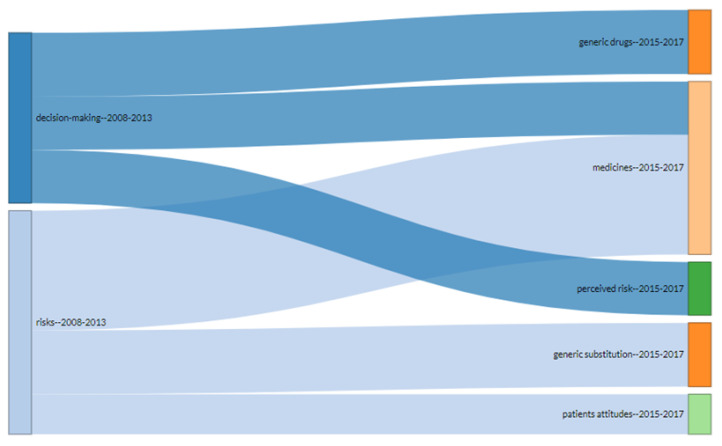
Thematic evolution of the research.

**Table 1 ijerph-18-04579-t001:** Search Strings applied to databases, and number of articles resulting from the search.

Search Strings of the Papers in the Corpus		
Database	Search String	Results
Web of Science	TS = (generic drug and purchase intention and attitude)	3
Science Direct	“generic drug” and “purchase intention” and “attitude”	8
Scopus	ALL (“generic drug” AND “purchase intention” AND “attitude”) AND (LIMIT-TO (DOCTYPE, “ar”)) AND (LIMIT-TO (LANGUAGE, “English”))	33
Lilacs	“generic drug” and “purchase intention” and “attitude”	1
Pubmed Central	((“generic drug”) AND “purchase intention”) AND “attitude”	0
Springer	purchase intention AND attitude AND “generic drug”	151
Embase	(‘generic drug’/exp OR ‘generic drug’) AND ‘purchase intention’ AND (‘attitude’/exp OR ‘attitude’)	2
TOTAL		198

**Table 2 ijerph-18-04579-t002:** Summary of studies from the textual corpus of the research.

Subject	*n*	Type	Focus	Main Findings	References
Consumers	Panel data drug utilization 1996–2001	Secondary Data	How the mix of consumer choices between generic and brand-name drugs might affect the average price of those brand-name drugs that are purchased.	We found that the average price paid by consumers for brand-name drugs falls substantially when generic script share rises.	[8]
Consumers	406	Questionnaire	Moderating role of an economic crisis in consumer purchase intention of generic drugs	Shopping attitude is positively influenced by both perceived quality and past experience. Role of doctors and pharmacist endorsers have no influence.	[13]
Consumers, Doctors, and Pharmacists	1069	Experimental study	Understand whether public health interventions in India are affecting the use of and access to generic drugs for non-communicable diseases.	Study generated evidence for health planners on how to optimize health services to improve access to medicines in a particular kind of setting.	[53]
Consumers	563	Secondary Data	How accounting information and message framing jointly impact consumer choice between brand name and generic drugs	Consumer choice between brand name and generic drugs, information on manufacturers’ profit margins, and costs in different frames evokes associations in memory, predisposing consumers to develop more or less favorable attitudes towards firms and their products.	[54]
Students	162	Questionnaire	To evaluate the effects of anthropomorphic images and information narration styles.	The search revealed that anthropomorphism of medications and narration styles could play a significant role in promotional messages for pharmaceuticals.	[55]
Government Officials and Pharmaceutical Executives	10 interviews	Interviews	Evaluate through the stakeholders of the pharmaceutical industry the determinants of cardiovascular generic drugs.	Whilst interviewees suggested that government policy plays an important role in shaping the industry, a significant force for change was ascribed to patient-derived factors.	[56]
Consumers	2222	Survey	Consumers’ purchase intention of generic drugs based on the theoretical framework of the theory of planned behavior	Positive effects of attitude, subjective norm, and past behavior on generic drugs purchase intention, while there are no positive effects on perceived behavioral control. Risk, trust in the pharmacist, brand sensitivity, and self-identity explained consumers’ intention to buy generic drugs.	[29]
Doctors, Pharmacists, and Consumers/Patients	58 Papers	Review article	Systematic search of published studies that focuses on physician, pharmacist, and patient/consumer perspectives.	A key factor in improving the confidence of these cohorts is the provision of information and education, particularly in the areas of equivalency, regulation, and in dispelling myths about generic medicines. Improving opinions regarding generics within the physician cohort may be of critical importance to improve the usage and acceptance of generic medicines in the future.	[57]
Patients	218	Questionnaire	Substitution of branded drugs by generics	Participative decision-making has no impact on purchase intention of generics, while perceived risk and price consciousness have a significant impact.	[58]
Consumers/Patients, Doctors, and Pharmacists	542	Questionnaire	To study the causal relationships influencing consumer purchase intention, including perceived risk, experience, and information provided by a physician and pharmacist as antecedents.	We found that the greater the perceived risk, the lower the motivation to request generic drugs, an effect that is reduced by the positive effect of experience.	[59]
Stockists, Doctors, and Consumers	155	Questionnaire	To evaluate the possibility of marketing specific low-cost drugs.	Cost of medicine is a significant factor influencing patients’ attitude towards the consumption of medicine.	[60]
Consumers	1278	Questionnaire	Influence of diseases on the level of agreement of prescription of generic drugs and sociodemographic factors.	Beliefs about the use of generic medicines are associated with the nature of the illness for which they are prescribed.	[30]
Opinion-leaders, Consumers/Patients	150	Questionnaire	Consumers, doctors, and pharmacists’ perception of generic drugs	Multiple factors may contribute to the decision to buy a generic drug. Price seems to be an important factor as well as effectiveness, safety, and trust.	[61]

**Table 3 ijerph-18-04579-t003:** Journals that form the research corpus.

Sources	Articles	SJR Best Quartil
International Journal of Pharmaceutical and Healthcare Marketing	2	Q3
Accounting Organizations and Society	1	Q1
BMC Health Services Research	1	Q1
BMC Medicine	1	Q1
Cadernos de Saúde Publica	1	Q2
Drug Information Journal	1	Q1
Health Policy	1	Q1
International Journal of Health Care Finance & Economics	1	Q1
Journal of Pharmaceutical Policy and Practice	1	Q1
Journal of Retailing and Consumer Services	1	Q1
Pharmacy World & Science	1	-
Research in Social & Administrative Pharmacy	1	Q1

**Table 4 ijerph-18-04579-t004:** Cluster with co-citations of references.

Clusters	Authors	Year	Journal	Total Link Strength	Citations
Cluster 1–Consumer attitude and behavior	Himmel W.	2005	Int. J. Clin. Pharm. Th.	41	6
	Quintal C.	2012	Health Policy	39	5
	Kobayashi E.	2011	Health Policy	33	4
	Kohli E.	2013	South Med. J.	27	3
	Patel A.	2010	Health Policy	27	3
	Shrank W.H.	2009	Med. Care	23	5
	Shrank W.H.	2011	Ann. Pharmacother.	22	5
Cluster 2–Views of patients and health professionals	Shrank W.H.	2009	Health Affair	28	4
	Babar Z.	2010	Pharm. World Sci.	27	3
	Hassali M.	2009	Int. J. Clin. Pharm. Pract.	27	3
	Kjoenniksen I.	2006	Pharm. World Sci.	23	3
Cluster 3–Risks associated with generic drugs	Heikkila R.	2007	Health Policy	23	3
	Ganther J.M.	2000	J. Am. Pharm. Assoc.	19	4

## Data Availability

All relevant data sets in this study are described in the manuscript.

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
