# Peer review of "Attitude and Purchase Intention to Generic Drugs"

_ijerph, 2021, doi:10.3390/ijerph18094579_

Round 1

Reviewer 1 Report

Comments for Authors: MS # ijerph-1126338 Title: Attitudes and intention to purchase generic drugs: Insights from a systematic mapping of literature

Thank you for the opportunity to read this literature review paper.  There are some interesting insights.  The paper is well written.

Here are two suggestions for improving the paper.  The first is around cluster one or the perceived risk.  Since the authors suggest that the paper includes intentions to purchase, a more profound discussion around perceptions of risk and unintended biases is needed.  Risk aversion and its implication on purchasing is rich in the marketing literature and should be acknowledged in the paper.

The second suggestion is a limitation does seem to exist around the locale of published papers.  Figure 5 and the geographic distribution of published papers seem an explicit limitation to the generalizability of any conclusions drawn. 

Author Response

Thank you so much for reviewing our manuscript entitled “Attitudes and intention to purchase generic drugs: Insights from a mapping of literature (Manuscript ID: ijerph-1126338) and inviting us to revise and resubmit our manuscript to International Journal of Environmental Research and Public Health. Based on the reviewer’s suggestions, we thoroughly revised our manuscript and believe the manuscript is significantly improved.

We are grateful to you and the reviewers for the positive feedback and constructive comments. We have organized this response letter around all the comments we received. Following the letter narrative, we provided responses outlining the detailed changes made to the manuscript. We believe the revised paper is a considerable improvement after addressing all the feedback from you and the reviewers. We have made every effort to answer all the comments we received, and we hope the manuscript is now suitable for publication in International Journal of Environmental Research and Public Health. Please feel free to contact us with any questions. We greatly appreciate your kind consideration of our manuscript, and look forward to your decision. Thank you so much!

Sincerely,

Ricardo, Cássia, Wesley and Claudimar

#Review 1

Thank you for the opportunity to read this literature review paper.  There are some interesting insights.  The paper is well written.

 Response: Thank you very much for your kind words! We really appreciate it.

Here are two suggestions for improving the paper.  The first is around cluster one or the perceived risk.  Since the authors suggest that the paper includes intentions to purchase, a more profound discussion around perceptions of risk and unintended biases is needed.  Risk aversion and its implication on purchasing is rich in the marketing literature and should be acknowledged in the paper.

Response: Thank you very much for the excellent comment and observation. We appreciate the two suggestions made. We have now brought a more in-depth discussion of risk perceptions and unintended purchasing biases. Indeed, risk aversion and its implications for purchases is an extensive topic in the marketing literature. We are grateful to you for that.

Studies that evaluate perceived risk show that this is a multidimensional construct involving psychological risk, time risk, physical risk, and social risk [24]. Physicians and patients perceive risk in the use of generic drugs because they consider that they are not as safe or effective as the brand name alternatives [25]. Clinical effectiveness was reported by 71.9% of physicians in Saudi Arabia as the most influential factor affecting the prescription of brands over local generic medication [26]. On the other hand, shopping attitude and purchase intentions are sensitive to an economic recession. The perceived risk in relation to the use of generic drugs was reduced during periods of economic crisis in Brazil

The second suggestion is a limitation does seem to exist around the locale of published papers.  Figure 5 and the geographic distribution of published papers seem an explicit limitation to the generalizability of any conclusions drawn. 

Response: Thank you very much for this observation. You are right about explicitly limiting the generalization of any conclusions drawn. This made us reflect on the discussions related to Figure 5, and we concluded that it does not add value to the manuscript. In this way, we think it is better to exclude it from the text. We are grateful to you for this advice.

The implementation of specific regulations to encourage the production and use of generic drugs around the world occurred at different times and in different ways [67]. Likewise, interest in academic research related to the intention to purchase generic drugs also varied from country to country. Based on the textual corpus, India and the United States are considered mature pharmaceutical markets based on the time since sec-tor-specific regulation was implemented [68]. In the USA, 9 out of 10 prescriptions filled out by doctors are for generic drugs, increasing market competition and the availability of generic drugs [69]. In turn, India stands out as the largest global supplier of generic drugs, with 40% of demand directed to the United States consumer market [70]. The three-field plot analysis in Figure 5 highlights the differences between the two countries in the topics of interest regarding generic drugs.

Figure 5 correlates the countries of origin of the research with the information from the abstracts and sources cited in a Sankey diagram [39]. The width of the arrows demonstrates the strength of the relationship between the three variables that were surveyed. Research on intention to purchase generic drugs in the United States shows a strong correlation with the word’s generic drugs, consumers, and prescription, reflecting the country's concern with its consumer pharmaceutical market and buying behavior and the prescription of generic drugs [71]. The key periodicals are the Journal of Consumer Research, Medical Care, the Journal of Marketing, and the Journal of Applied Psychology.

Reviewer 2 Report

Comments to "Attitudes and intention to purchase generic drugs: Insights from a systematic mapping of literature":

- Abstract:
. "In recent decades" Which ones? What period does it cover?
. It must include the number of documents and period

- Write in a non-personal way. Avoid we, our, me, I

Introduction
- It's very brief. How have you detected the key terms? What literature has treated them? Why that period and not another?
- It must detail the structure of the document
- What are the hypotheses?
Material and methods
"The This study"?
Why do you use this methodology? Is it systematic or bibliometric?

Discussion
- You should discuss the topics for periods with the literature that you should have reviewed.

Conclusion
- You should conclude with the objective and hypothesis

Author Response

Thank you so much for reviewing our manuscript entitled “Attitudes and intention to purchase generic drugs: Insights from a mapping of literature (Manuscript ID: ijerph-1126338) and inviting us to revise and resubmit our manuscript to International Journal of Environmental Research and Public Health. Based on the reviewer’s suggestions, we thoroughly revised our manuscript and believe the manuscript is significantly improved.

We are grateful to you and the reviewers for the positive feedback and constructive comments. We have organized this response letter around all the comments we received. Following the letter narrative, we provided responses outlining the detailed changes made to the manuscript. We believe the revised paper is a considerable improvement after addressing all the feedback from you and the reviewers. We have made every effort to answer all the comments we received, and we hope the manuscript is now suitable for publication in International Journal of Environmental Research and Public Health. Please feel free to contact us with any questions. We greatly appreciate your kind consideration of our manuscript, and look forward to your decision. Thank you so much!

Sincerely,

Ricardo, Cássia, Wesley and Claudimar

Comments to "Attitudes and intention to purchase generic drugs: Insights from a systematic mapping of literature":

- Abstract:
. "In recent decades" Which ones? What period does it cover?
. It must include the number of documents and period

Response: Thank you very much for reading our article and for the time dedicated to review. We are grateful to you for that. Faced with your doubt, we think it is better to rewrite the abstract. Regarding the period, it is shown in Figure 3, with emphasis on the growth in the number of publications since 1984.

The abstract is now as follows:

Generic drugs were instituted in 1984 in the United States. Since that time, many studies have been conducted in several countries into consumer attitude and behavior when purchasing generic drugs. Understanding the factors that can influence attitude and purchasing intention in this segment has been a challenge. Thus, this paper aims to present a mapping of the literature on the attitude toward and intention to purchase generic drugs and capture insights that can help define and improve promotional strategies for the use of these products. To identify articles related to the theme, we selected the Web of Science, Science Direct, Scopus, Lilacs, Pubmed Central, Springer and Embase databases time limited to June 2020, using the keywords "generic drug," "purchase intention and “attitude”. The results indicate that this topic is relatively new, with publications in the leading journals in the area, demonstrating its importance. Analysis revealed five strategic insights and showed that the research theme could be grouped into three clusters: (i) Consumer attitude and behavior, (ii) Perspective of patients and health professionals, and (iii) Assessment of the risks associated with generic medications to determine which factors can influence purchase intention, providing decision makers with a broader view with regard to directing public policy strategies in healthcare.

 Write in a non-personal way. Avoid we, our, me, I.

Response: Thank you for this observation. The text was revised, with appropriate language, now written in the third person.

Introduction
- It's very brief. How have you detected the key terms? What literature has treated them? Why that period and not another?

Response: Thank you very much for your comments. We included in the Introduction, the existing literature and related to the keywords used in research for the textual corpus. Regarding the choice of the period, we used the methodology without delimiting the time, with articles published until the date of the research in June 2020.

- It must detail the structure of the document
- What are the hypotheses?

Response: Thank you for this excellent suggestion. We improved the structure of the manuscript. On the other hand, we did not include hypotheses because it is not a meta-analytic study and because it was not the objective of this paper. Anyway, we have included in the suggestion of future studies to test the hypotheses through meta-analytic research.

Material and methods
"The This study"?

Response: Thank you for this observation. We apologize for this error. We made the correction in the text, as indicated by you.

Why do you use this methodology? Is it systematic or bibliometric?

Response:  Thank you for this question. To systematically review and get an overview of studies in a given research area, Systematic Literature Review (SLR) and Systematic (Literature) Mapping (SLM or SM) studies are the established approaches.  According to Petersen et al. (2008), a systematic mapping (SM) is a method to review, classify, and structure papers related to a specific research.

In the literature, according to Fagundes et al. (2020), Bibliometric methods introduce a measure of objectivity in scientific assessment, attenuating bias and aggregating opinions of various researchers. On the other hand, Zupic and Cater (2015) and Tranfield et al. (2003) proposed a five-phase methodology for critical literature mapping, promoting the identification of gaps and the generation of research insights in a given area of knowledge: 1. definition of the research design and method; 2. compilation of bibliometric data; 3. data analysis; 4. visualization of results; and 5. interpretation of results. In this article, we adopted a similar systematic literature mapping process to analyze "attitudes and intention to purchase generic drugs and bring Insights of the mapping of literature.

 Petersen, K., Feldt, R., Mujtaba, S., Mattsson, M. (2008). Systematic mapping studies in software engineering, in: 12th International Conference on Evaluation and Assessment in Software Engineering (EASE), 71–80.

Vahid Garousi, Ali Mesbah, Aysu Betin-Can, Shabnam Mirshokraie. (2013). A systematic mapping study of web application testing, Information and Software Technology, 55(8), 1374-1396.

Zupic, I., & Cater, T. (2015). Bibliometric methods in management and organization. Organizational Research Methods, 18(3), 429–472.http://dx.doi.org/10.1177/1094428114562629

Tranfield, D; Denyer, D; Smart, P Towards a methodology for developing evidence-informed management knowledge by means of systematic review British Journal of Management. 2003; 14:207-222.

Discussion
- You should discuss the topics for periods with the literature that you should have reviewed.

Response: Thank you very much for this comment. We really appreciate that. The textual corpus indicates the articles published by period. We also inserted Table 1. In the article, we analyzed discussions of the topics taking into account the research carried out before 2013 and after this year. In this context, we evaluated the evolution of consumer attitude and intention to purchase generic drugs over two different time periods due to the fact that there has been an increase in the number of publications from 2013.

“As shown in Figure 6, the publications before 2013 can be divided into two clusters: decision-making and risks. After 2013, the decision-making cluster can be segmented into generic drugs, medicines and perceived risks, demonstrating that decision making is more focused on the product through costs and profit margin [13,41], promotional materials [42], public policies [43] and quality of generic drugs [44]. Likewise, the risks cluster can be divided into medicines, generic substitution and patient attitude, which may suggest that patients are suspicious regarding the effectiveness of generic drugs, reinforcing the need to improve communication to health professionals and final consumers [45] and to facilitate access to generic drugs [40]”.

Conclusion
- You should conclude with the objective and hypothesis

Thank you for the comment.  In the first paragraph of the conclusion, we have included some adjustments where it reveals more clearly the proposed objective. As for the hypotheses explained earlier, this study does not deal with metanalysis; for this reason, we did not bring any hypotheses. On the other hand, the idea is valid; We highlight the need for studies with hypothesis validation in studies that report to the attitude and intention to purchase generic drugs.

Reviewer 3 Report

Overall Feedback:

The authors provide a systematic bibliographic analysis about research on the attitudes and intentions to purchase generic drugs. The authors provide a clear methodology of selection criteria and analysis processes, including literature citations in the selected body of literature as well as citations of the selected body of research. The topic is interesting and relevant from a research perspective, due to its great social and economic relevance.

From my personal point of view the study lacks more actionable insights and a more formal integration of the existing literature into a coherent theoretical framework on the attitudes and purchase intentions for generic drugs.

Overall, I think this study misses a discussion of the empirical results of the analyzed research corpus. A table investigating significant and insignificant empirical findings from the 13 studies would be more than welcome to enhance the insights of the present research. Based on this, Independent and dependent variables that impact purchase intentions could be grouped, and based on the empirical findings might even be prioritized. This would facilitate that a reader could truly use the present study as a summary instead of analyzing the research corpus again to get a well-founded opinion on the research in this field.

Such a theoretical framework would also bring along better guidance for future research and pinpoint towards pressuring issues in the field by highlighting knowledge gaps.  A deeper assessment of the research corpus in terms of theoretical backbone as well as empirical findings is mandatory for this article in order to warrant publication.

The topic is interesting and relevant, the methods of selection and analysis seem sound, and the article is well written. The missing part for me is the intellectual contribution of analyzing empirical findings (by highlighting them, or, if possible via a meta-analysis) and combining them into a theoretical framework in order to derive more actionable insights

Specific Comments

The analysis around figure 4 is not completely clear to me. Are those the references most cited in the selected body of literature, or are those the interlinkages in the research corpus itself? The journals of the figure and the journals of the research corpus do not coincide, so I assume the former, yet both sets of studies include 13 studies adding to my confusion. Then again, the identified clusters seem to refer to the research corpus, so I simply do not get it. More clarification is necessary here, and if there are potential mistakes or inconsistencies those should be corrected (it is fully possible though that I simply misunderstood the analysis here).

Insights: The third insight is not fully justifiable. It is purely based on word counts, yet the occurrence of words in scientific research does not imply that the topics are important, significant or relevant for practice.

The forth insight is not clear to me. I did not understand how it was derived from the analysis.

The insights section does not convince me.

On the one hand, sentences are rather broad and vague and might not require a literature review, in order to be written up. On the other hand such statements do not generate any actionable insight. E.g.:

 “This insight shows that generic regulation and pricing policies vary from country to country and their implementation presents different challenges and opportunities.”

On the other hand, other statements make claims that are probably not to be fully backed by the results: E.g.:

“Likewise, the risks cluster can be divided into (i) Medicines, (ii) Generic substitution and (iii) Patients’ attitudes, which may suggest that patients are suspicious about the effectiveness of generic drugs. This insight suggests the need for improvements to the communication strategy between doctors, health professionals and the final consumer.”

First of all, I do not see any reason why Medicines, Substitution and Attitudes would imply any suspicion. The causal linkage here seems rather far-fetched. Furthermore, to claim that these three clusters demand a better communication strategy does not necessary follow from this.

Minor comments:

Add Market share to the dominating players if possible

Figure 1: close the parenthesis.

  1. Discussion: cut the word “Authors” in the beginning

I wish the authors the best of luck with this research.

Author Response

Thank you so much for reviewing our manuscript entitled “Attitudes and intention to purchase generic drugs: Insights from a mapping of literature (Manuscript ID: ijerph-1126338) and inviting us to revise and resubmit our manuscript to International Journal of Environmental Research and Public Health. Based on the reviewer’s suggestions, we thoroughly revised our manuscript and believe the manuscript is significantly improved.

We are grateful to you and the reviewers for the positive feedback and constructive comments. We have organized this response letter around all the comments we received. Following the letter narrative, we provided responses outlining the detailed changes made to the manuscript. We believe the revised paper is a considerable improvement after addressing all the feedback from you and the reviewers. We have made every effort to answer all the comments we received, and we hope the manuscript is now suitable for publication in International Journal of Environmental Research and Public Health. Please feel free to contact us with any questions. We greatly appreciate your kind consideration of our manuscript, and look forward to your decision. Thank you so much!

Sincerely,

Ricardo, Cássia, Wesley and Claudimar

Overall Feedback:

The authors provide a systematic bibliographic analysis about research on the attitudes and intentions to purchase generic drugs. The authors provide a clear methodology of selection criteria and analysis processes, including literature citations in the selected body of literature as well as citations of the selected body of research. The topic is interesting and relevant from a research perspective, due to its great social and economic relevance.

Response:  Response: Thank you very much for your kind words! We really appreciate it.

From my personal point of view the study lacks more actionable insights and a more formal integration of the existing literature into a coherent theoretical framework on the attitudes and purchase intentions for generic drugs.

Response: Thank you very much for the suggestion. We rewrote part of the introduction with emphasis on purchase intent and perceived risk.

Ferreira et al. (2017) highlights that the attitude towards purchasing generic drugs can be influenced by perceived quality, product attributes, past experience, and doctors’ recommendations. [17] show that product attributes can influence purchase intention. In this context, the conclusion can be reached that understanding the consumer's journey in the purchase of generic medication related to attitude and purchase intention has been a challenge in the literature.

“To define purchase intention, this study was based on the definition proposed by [18], [19] and [20]. These authors highlight that purchase intention is classified as one of the components of consumers' cognitive behavior [18] regarding the means, form, and probability of buying a specific product [18]; [21]. This study was based on research conducted by [22]. This author explains that purchase attitude is based on determining the behavior for deciding on a future purchase. In this context, [23] highlight that purchase attitude includes trust in the product and familiarity related to the consumer's purchasing power. To this author, in the minds of consumers, the buying attitude is based on the result of (i) cognitive, (ii) emotional, and (iii) affective reactions. On the other hand, studies that evaluate perceived risk show that this is a multidimensional construct involving psychological risk, time risk, physical risk, and social risk [24]’.

“Studies that evaluate perceived risk show that this is a multidimensional construct involving psychological risk, time risk, physical risk, and social risk [24]. Physicians and patients perceive risk in the use of generic drugs because they consider that they are not as safe or effective as the brand name alternatives [25]. Clinical effectiveness was reported by 71.9% of physicians in Saudi Arabia as the most influential factor affecting the pre-scription of brands over local generic medication [26]. On the other hand, shopping at-titude and purchase intentions are sensitive to an economic recession. The perceived risk in relation to the use of generic drugs was reduced during periods of economic crisis in Brazil [13]”.

Overall, I think this study misses a discussion of the empirical results of the analyzed research corpus. A table investigating significant and insignificant empirical findings from the 13 studies would be more than welcome to enhance the insights of the present research. Based on this, Independent and dependent variables that impact purchase intentions could be grouped, and based on the empirical findings might even be prioritized. This would facilitate that a reader could truly use the present study as a summary instead of analyzing the research corpus again to get a well-founded opinion on the research in this field.

Response: Thank you for this important observation and comment. To facilitate understanding for readers. we made Table 1 with all the articles to facilitate the visualization of the articles that make up the research corpus. This suggestion is excellent and we appreciate it.

Such a theoretical framework would also bring along better guidance for future research and pinpoint towards pressuring issues in the field by highlighting knowledge gaps.  A deeper assessment of the research corpus in terms of theoretical backbone as well as empirical findings is mandatory for this article in order to warrant publication.

Response: Response: Thank you very much for this observation. When building Table 1, we made a deeper assessment of the research corpus in terms of theoretical structure as well as empirical findings.

The topic is interesting and relevant, the methods of selection and analysis seem sound, and the article is well written.

Response: Thank you very much for your kind words! We really appreciate it.

The missing part for me is the intellectual contribution of analyzing empirical findings (by highlighting them, or, if possible, via a meta-analysis) and combining them into a theoretical framework in order to derive more actionable insights

Response: Thank you very much for this suggestion. Your suggestion is in line with our research proposal regarding relevance, application of selection methods, and textual corpus analysis. In this sense, the work complied with what portrays the research problem. On the other hand, we highlight the recommendations for elaborating future works to accomplish a meta-analysis as a complement to this work, where some statistics can be estimated as an analysis of variance. Also, it aims to test hypotheses and assess the behavior of the variance of the results, used as factors common to articles in the corpus.

Specific Comments

The analysis around figure 4 is not completely clear to me. Are those the references most cited in the selected body of literature, or are those the interlinkages in the research corpus itself? The journals of the figure and the journals of the research corpus do not coincide, so I assume the former, yet both sets of studies include 13 studies adding to my confusion. Then again, the identified clusters seem to refer to the research corpus, so I simply do not get it. More clarification is necessary here, and if there are potential mistakes or inconsistencies those should be corrected (it is fully possible though that I simply misunderstood the analysis here).

Thank you very much for bringing this discussion. We rewrote some analysis and made it clearer in the text what Figure 4 actually expresses. The same Figure shows 3 clusters of the relationship network between the cited references of the articles that make up the research corpus. We believe that the interpretation is now more consistent. Figure 4 shows the co-citation networks with the references with the highest interconnection of the research corpus.

Insights: The third insight is not fully justifiable. It is purely based on word counts, yet the occurrence of words in scientific research does not imply that the topics are important, significant or relevant for practice.

Response:  Concerning the co-citation network of references, it shows the validity of the third bibliometric law of Zipf. The clusters were identified and deeply evaluated based on the occurrences of the words mentioned in the abstracts.The purpose is to support decision-makers and researchers in identifying themes mainly addressed in this corpus's articles by identifying sub-themes. Notably, the extracted network's relevant topics seek to portray what the authors described in their scientific papers through abstracts, with peer validation.

The forth insight is not clear to me. I did not understand how it was derived from the analysis.

Response: Thank you for your comments. The fourth insight is related to implementing specific regulations corroborated by Chawla et al. (2014). We derive this relationship due to the most present research in mature markets where regulation becomes more detailed.

The insights section does not convince me.

Response:  Thanks for your comment. We restructured the insights section. The first insight is related to the concentration of the studies. The second insight was related to the frequency of citations of articles since 1980. Given the relevance of the topic, it reflects the importance of prior knowledge for the dissemination of research. The fourth insight is related to the implementation of specific regulations to encourage the production and use of generic drugs. The review showed that the implementation of such regulations occurred around the globe at different times and in different ways. This insight also highlighted that these two countries are considered mature pharmaceutical markets due to the elapsed time since sector-specific regulation was implemented. The fifth insight shed light on research on generic drug purchase intent in India (see new Figure 5). Research on intention to purchase generic drugs in this country has the greatest correlation with the words ‘health’ and ‘medicines’. According to previous research, India is an important producer of generic medicines, but its population has difficulty in accessing medicines, especially for chronic diseases.

The sixth insight was related to the evolution of the central theme using two distinct time periods, before and after 2013, based on the increase in the number of publications subsequent to this date.On the one hand, sentences are rather broad and vague and might not require a literature review, in order to be written up. On the other hand such statements do not generate any actionable insight. E.g.: “This insight shows that generic regulation and pricing policies vary from country to country and their implementation presents different challenges and opportunities.”

Response: Thank you very much for this comment. We corrected this part in the text.

On the other hand, other statements make claims that are probably not to be fully backed by the results: E.g.:

“Likewise, the risks cluster can be divided into (i) Medicines, (ii) Generic substitution and (iii) Patients’ attitudes, which may suggest that patients are suspicious about the effectiveness of generic drugs. This insight suggests the need for improvements to the communication strategy between doctors, health professionals and the final consumer.”

First of all, I do not see any reason why Medicines, Substitution and Attitudes would imply any suspicion. The causal linkage here seems rather far-fetched. Furthermore, to claim that these three clusters demand a better communication strategy does not necessary follow from this.

Response:  Thank you very much for this suggestion. This insight really got confused. In order not to compromise the robustness of the work, we have eliminated the sixth insight. We are grateful to you for that.

Minor comments:

Add Market share to the dominating players if possible

Response: Excellent suggestion. Added market share of the companies mentioned. We take the Global Generic Drugs Market (2020) report as a basis.

Figure 1: close the parenthesis.

Response: We apologize for this error. We corrected it in the text.

I wish the authors the best of luck with this research.

Response: Thank you very much for your kind words! We really appreciate it. We are very grateful for your attention in reading our work and for all suggested improvement points. Although we have submitted several academic papers for renowned journals over time, we can say with certainty that it was the best review we have ever received. We believe that by addressing your comments and suggestions, our revised manuscript is significantly improved. Receive our sincere thanks.

Reviewer 4 Report

My congratulations to the authors for their manuscript entitled “Attitudes and intention to purchase generic drugs: Insights from a systematic mapping of literature”.

My only recommendation would be to include other related studies to detect systematic areas of literature. This would help raise the justification for this research. It could already be under the same programs VOSviewer and Biblioshiny-Bibliometrix.

Author Response

Thank you so much for reviewing our manuscript entitled “Attitudes and intention to purchase generic drugs: Insights from a mapping of literature (Manuscript ID: ijerph-1126338) and inviting us to revise and resubmit our manuscript to International Journal of Environmental Research and Public Health. Based on the reviewer’s suggestions, we thoroughly revised our manuscript and believe the manuscript is significantly improved.

We are grateful to you and the reviewers for the positive feedback and constructive comments. We have organized this response letter around all the comments we received. Following the letter narrative, we provided responses outlining the detailed changes made to the manuscript. We believe the revised paper is a considerable improvement after addressing all the feedback from you and the reviewers. We have made every effort to answer all the comments we received, and we hope the manuscript is now suitable for publication in International Journal of Environmental Research and Public Health. Please feel free to contact us with any questions. We greatly appreciate your kind consideration of our manuscript, and look forward to your decision. Thank you so much!

Sincerely,

Ricardo, Cássia, Wesley and Claudimar

My congratulations to the authors for their manuscript entitled “Attitudes and intention to purchase generic drugs: Insights from a systematic mapping of literature”.

Response: Thank you very much for your kind words! We really appreciate it.

My only recommendation would be to include other related studies to detect systematic areas of literature. This would help raise the justification for this research. It could already be under the same programs VOSviewer and Biblioshiny-Bibliometrix.

Response: Thank you for this excellent suggestion. We are grateful to you. To meet this suggestion, we have inserted a future recommendation in order to better understand the consumer who is willing to buy generic drugs.

Round 2

Reviewer 2 Report

Comments to the manuscript "Attitudes and intention to purchase generic drugs: Insights from a systematic mapping of literature":

- The literature review is still scarce (Intro section) to be able to deduce a gap that leads to this manuscript. Nor is it adequately reflected which is this.
- What is the sense of analyzing the citations in a systematic review? Is it not more logical to study the lines or themes that the selected documents study?
- What is the main contribution of this study to the research field? Why should a reader read it or publish it in the journal?
- You should carry out a greater review of documents to clarify the objective and define the hypotheses

Author Response

Dear Editor and Reviewer

Thank you so much for reviewing our manuscript entitled “Attitudes and intention to purchase generic drugs: Insights from a systematic mapping of literature (Manuscript ID: ijerph-1126338) and inviting us to revise and resubmit our manuscript to International Journal of Environmental Research and Public Health. Based on the reviewer’s suggestions, we thoroughly revised our manuscript and believe the manuscript is significantly improved.

We are grateful to you and the reviewer for the positive feedback and constructive comments. We have organized this response letter around all the comments we received. Following the letter narrative, we provided responses outlining the detailed changes made to the manuscript. We believe the revised paper is a considerable improvement after addressing all the feedback from you and the reviewer. We have made every effort to answer all the comments we received, and we hope the manuscript is now suitable for publication in International Journal of Environmental Research and Public Health. Please feel free to contact us with any questions. We greatly appreciate your kind consideration of our manuscript and look forward to your decision. Thank you so much!

Sincerely,

Ricardo, Cássia, Wesley and Claudimar

Response to reviewer

---------------------------

Comments to the manuscript "Attitudes and intention to purchase generic drugs: Insights from a systematic mapping of literature":

- The literature review is still scarce (Intro section) to be able to deduce a gap that leads to this manuscript. Nor is it adequately reflected which is this.

Thank you very much for your comments. We corrected the mistake. To solve the problem related to the scarcity of studies to deduce the research gap, we reviewed the entire introduction of the article. Changes and corrections are highlighted in the main text, in red. Based on your suggestions, we believe that the article improved by making the research objective and gap clearer. We are grateful to you for that.

List of new articles evaluated that helped to improve the introduction, definition of the GAP, and research objective:

  1. Ganther, J M; Kreling, D H Consumer perceptions of risk and required cost savings for generic prescription drugs Journal of the American Pharmaceutical Association. 2000; 40:378-383.
  2. Hughes, J W; Moore, M J; Snyder, E A "Napsterizing" Pharmaceuticals: Access, Innovation, and Consumer Welfare National Bureau of Economic Research 2002; 9229.
  3. Sewell, K; Andreae, S; Luke, E; Safford, M M Perceptions of and Barriers to Use of Generic Medications in a Rural African American Population, Alabama, 2011 Preventing Chronic Disease 2012; 9:142.
  4. Muzumdar, J M; Schommer, J C; Hadsall, R S; Huh, J Effects of anthropomorphic images and narration styles in promotional messages for generic prescription drugs Research Social and Administrative Pharmacy 2013; 9:60-79.
  5. Figueiras, M. J.; Marcelino, D.; Cortes, M. A. People’s view on the level of agreement of generic medicines for different illnesses Pharmacy, World and Science. 2008; 30:590-594.
  6. Fraeyman, J; Peeters, L; Van Hal, G; Beutels, P; De Meyer, G R Y; Loof, H Consumer Choice Between Common Generic and Brand Medicines in a Country with a Small Generic Market Journal of Managed Care Specialty Pharmacy 2015; 21(4):288-296.
  7. Kobayashi, E; Karigome, H; Sakurada, T; Satoh, N; Ueda, S Patients’ attitudes towards generic drug substitution in Japan Health Policy 2011; 99:60-65.
  8. Gillbody, S; Wilson, P; Watt, I Benefits and harms of direct to consumer advertising: a systematic review BMJ Quality & Safety 2005; 14:246-250.
  9. Rida, N A; Zaidan, M; Ibrahim, M I M. An exploratory insight on pharmaceutical sector and pricing policies in Qatar Global Journal of Pharmacy & Pharmaceutical Science 2017; 1:4.
  10. Pruszydlo, M G; Walk-Fritz, S U; Hoppe-Tichy, T; Kaltschmidht, J; Haefeli, W E Development and evaluation of a computerised clinical decision support system for switching drugs at the interface between primary and tertiary care BMC Medical Informatics and Decision Making 2012; 12:137.
  11. Dylst, P; Vulto, A; Godman, B; Simoens, S Generic medicines: Solutions for a sustainable drug market? Applied Health Economics and Health Policy 2013; 11:437-443.
  12. Fischer KE, Stargardt T. The diffusion of generics after patent expiry in Germany. Eur J Health Econ. 2016;17(8):1027–40.
  13. Howard, J N; Harris, I; Frank, G; Kiptanui, Z; Qian, J; Hansen. R. Influencers of generic drug utilization: A systematic review Research in Social and Administrative Pharmacy. 2018; 14: 619-627.
  14. S. Government Accountability Office. Drug pricing: research on savings from generic drug use.http://www.gao.gov/assets/590/588064.pdf; 2012. Accessed17.05.07.
  15. US Department of Health and Human Services. Guidance for industry: ANDAsubmissions - Amendments and easily correctable deficiencies under GDUFA.https://www.fda.gov/downloads/Drugs/GuidanceComplianceRegulatoryInformation/Guidances/UCM404440.pdf; 2014. Accessed 17.03.2021.
  16. Hassali, M. A., Alrasheedy, A. A., Aljadhey, H. The experiences of implementing generic medicine policy in eight countries: A review and recommendations for a successful promotion of generic medicine use. Saudi Pharmaceutical Journal. 2014; 22(6), p. 491-503.
  17. Tuncay, B.; Pagano, S.; De Santis, M.; Cavallo, P. Prescribing Behavior of General Practitioners for Generic Drugs. Int. J. Environ. Res. Public Health 2020, 17, 5919. https://doi.org/10.3390/ijerph17165919.

- What is the sense of analyzing the citations in a systematic review? Is it not more logical to study the lines or themes that the selected documents study?

Response: Thank you for the comment. The meaning of using the networks from the co-citation analysis is in the fact of discovering which documents are being cited together based on the evaluated texts of our research corpus. Zupic and Cater, 2013 address this theme, showing that the co-citation analysis is necessary to reflect the state of the art of a field of knowledge, visualizing the themes in which they correlate to build similarity measures. It is worth mentioning that numerous studies are using the same analysis seeking to identify trends in the most varied themes (Tamilmani, Rana, and Dwivedi, 2017; Klarin, 2019; Kacetl et al., 2020).

  1. Tamilmani, K., Rana, N.P. & Dwivedi, Y.K. Consumer Acceptance and Use of Information Technology: A Meta-Analytic Evaluation of UTAUT2. Inf Syst Front (2020). https://doi.org/10.1007/s10796-020-10007-6
  2. Klarin, A. Mapping product and service innovation: A bibliometric analysis and a typology Technological Forecasting and Social Change 149:119776.
  3. Kacetl, J., Maresová, P., Masjuriy, R., Selamat, A. Ethical Questions linked to rare diseases and orphan drugs – a systematic review Risk Manag Healthc Policy 13:2125-2148.

- What is the main contribution of this study to the research field? Why should a reader read it or publish it in the journal?

Response: Thank you very much for your comments. This study's main contributions are the methodological framework, the set of insights that can assist decision-makers in the field of generic drugs. Analysis revealed five strategic insights and showed that the research theme could be grouped into three clusters: (i) Consumer attitude and behavior, (ii) Perspective of patients and health professionals, and (iii) Assessment of the risks associated with generic medications to determine which factors can influence purchase intention, providing decision-makers with a broader view about directing public policy strategies in healthcare. This finding corroborates studies by (Howard et al. 2018) that addressed the use of generic drugs and reinforced the main areas of interest and need proactive strategies to assist in purchasing generic drugs.

From the discussions presented, we believe that this research can help with elaborating hypotheses for developing new quantitative studies to empirically verify the impact of the proposed recommendations, considering maximizing the use of generic drugs.

- You should carry out a greater review of documents to clarify the objective and define the hypotheses

Response: Thanks for the comment. We made adjustments to the research objective to clarify the purpose of the article. The suggestion to create hypotheses in the study was very much appreciated, and we thank you for that. In this study, it was not possible to include them because we did not analyze the textual corpus from a perspective of metallization. However, we have included the hypothesis proposition in the recommendation for future studies that explore the thematic attitude and purchase intention for generic drugs.

We are happy with your suggestions. Certainly, they were fundamental in improving our work. We are grateful to you for all the effort, time, and dedication in reading the article.